# Highly efficient exciton-exciton annihilation in single conjugated polymer chains

Nicola J. Fairbairn ⓘ , Olga Vodianova ⓘ , Bernhard V. K. J. Schmidt ⓘ &
Gordon J. Hedley ⓘ ✉

The number of excitons that conjugated polymers can support at any one time underpins their optoelectronic performance in light-emitting diodes and as laser gain media, as it sets a natural limit on exciton density. Here we have measured the time-resolved photon statistics of single, isolated chains of polyfluorene to extract the absolute number of independent emitting sites present and its time dependence by studying the intramolecular exciton-exciton annihilation. We find that after 100 ps each chain can only support 1 or 2 independent excitons, and that even at the earliest times this number rises only to 4, suggesting a high degree of electronic coupling between chromophores that facilitates efficient exciton-exciton annihilation. In circumstances where a low density of low-energy sites is present, annihilation between them still dominates. The results indicate that achieving high exciton densities in conjugated polymers is difficult, and in applications where it is desirable new strategies should be devised to control exciton-exciton annihilation.

Conjugated polymers have attractive optoelectronic properties for use in organic light-emitting diodes (OLEDs), photovoltaic cells and lasers[1–10]. Excited states are typically Frenkel excitons, delocalised coulombically-bound electron-hole pairs. These excitons hop via För-ster resonance energy transfer (FRET) initially downhill, then within the available thermal energy via diffusion[10,11]. How many excitons a polymer chain can support simultaneously, its exciton density, is important in defining its overall optoelectronic performance as it sets a limit on relevant device parameters, e.g. OLED brightness. Exciton motion via FRET enables two of them to come in close spatial proximity to each other, where they can undergo exciton-exciton annihilation (EEA)[12–16], a form of self-FRET that leads to the loss of one exciton per encounter event. Consequently, for mobile excitons, annihilation determines the upper limit of exciton density, and thus is important to understand and control in organic semiconducting devices, such as in an OLED operating under high current injection for high brightness, where the overall efficiency of the device decreases[17], or in an organic laser, where annihilation is a loss channel that increases thresholds and works against any attempt at electrical injection[18].

Measuring the absolute upper limit of allowable exciton densities in conjugated polymers and how this evolves during the excited state lifetime due to annihilation is inherently difficult. An indication that the density limit has locally been reached in some places is when annihilation is observed. This requires high photon fluences when under optical excitation i.e. a significant fraction of the excitons need to be annihilated for this to be measurable in photoluminescence or tran-sient absorption[12–16]. However, an underlying non-annihilating popu-lation typically remains that needs to be divided out, complicating analysis. This is exacerbated when considering that non-uniform laser excitation profiles (normally Gaussian) are used. Secondly, any attempt to determine the absolute exciton density requires assump-tions on the mass density and number of optically active chromo-phores in the sample volume that is being measured. The former is non-trivial to measure accurately, while the latter is subject to sig-nificant error.

Here we have used single-molecule spectroscopy[19–21] to study individual chains of poly(9,9′-dioctylfluorene) (PFO). By measuring the time-resolved photon statistics of fluorescence from single polymer chains for the first time, we can automatically monitor EEA in the very rare cases where it can happen, free from the much larger background of circumstances where it does not−in our work, this is at a rate of 35 per million emission events. We find that annihilation is always present as a loss channel. We observe that each individual PFO chain is com-prised of only ~4 independent emitting sites at the earliest times we

School of Chemistry, University of Glasgow, Glasgow G12 8QQ, UK. ✉e-mail: Gordon.Hedley@glasgow.ac.uk

can measure (~50 ps), suggesting that fast sub-picosecond evolution of the excited state significantly limits how many excitons a single chain can support. This number drops to ~2 within 100 ps, i.e. a single polymer chain can only support two excitons or less after this time. Exploiting low-energy sites in PFO as an analogy for exciton protection strategies (low-energy sites surrounded by protective higher-energy ones), we find that these low-energy sites can still annihilate with each other, with 5 independent sites becoming 2 after 1 ns. Our findings suggest that enabling high exciton densities in conjugated polymers will require careful control of energetic and conformational landscapes to enable applications such as high-brightness OLEDs or electrical injection organic lasers.

## Results

Poly(9,9'-dioctylfluorene), chemical structure inset Fig. 1a, is a prototypical conjugated polymer which has been widely studied owing to its desirable deep blue emission and high photoluminescence (PL) quantum yield[22–25]. Optical or electrical excitation of PFO leads to the formation of excitons delocalised across 4-5 monomeric units[26]. Single chains of PFO embedded in a PMMA matrix were measured in a nitrogen environment on a homebuilt confocal fluorescence microscope with single photon counting, Fig. 1b (see Methods for a full description of experimental and sample preparation methods). Fluorescence scan images over a region were recorded to identify single chains of PFO, before each chain was then measured for longer (up to 30 s) durations. Expected single-chain PFO behaviour is observed[27–33], including fast mono-exponential lifetimes, Fig. 1c, and intensity blinking (see Supplementary Information). When emitted photons are measured on two detectors with a 50:50 beamsplitter in a Hanbury Brown and Twiss arrangement[34] photon antibunching[35] is observed, Fig. 1d. Here the correlation events, $N$, between the two detector channels are recorded for different time lags of laser pulse

periods ($\Delta t$). The correlation between photons detected at their "true" time ($\Delta t = 0$) is calculated ($N_C$) along with artificial time lags ($\Delta t \neq 0$), ($N_L$). The ratio between these is used to determine the number of independent emitting sites, $n$, on the chain using Eq. 1[36].

$$n = \frac{1}{\left(1 - \frac{N_C}{N_L}\right)} \qquad (1)$$

For 2215 measured chains of PFO, $n$ is found to be $2.07 \pm 0.09$. Crucially, this is a time-averaged result, i.e. over the excited state lifetime, there are 2.07 active independent emitting sites present in each PFO chain, weighted to early time where more photons are emitted. It is then pertinent to ask how this will vary with time.

To explore any time dependence on the number of emitting sites, we then apply our recently developed time-resolved antibunching (TRAB) technique[37] to single polymer chains for the first time. To test that there is a response, we first applied three large filter windows, Fig. 2a, to the recorded photons, yielding separate antibunching results for each respective filter window, Fig. 2b. It is evident immediately that there is a change in the number of independent emitting sites on single chains of PFO between these time windows. At the earliest times (0–150 ps), ~2.7 emitting sites are present. In the next (160–360 ps), this drops to ~2.1, while in the final window (370–870 ps), this drops further to ~1.7. We have adjusted the lengths of the windows to ensure there are similar numbers of correlation events (~4000 on lateral peaks) in each, and thus similar signal-to-noise ratios.

Next, rather than looking at defined large windows, we can instead calculate a continuous measure of the ratio between the central and lateral values ($N_C/N_L$), with a step size of 20 ps, as shown in Fig. 2c, for 2215 individually measured chains. This can then be converted into the

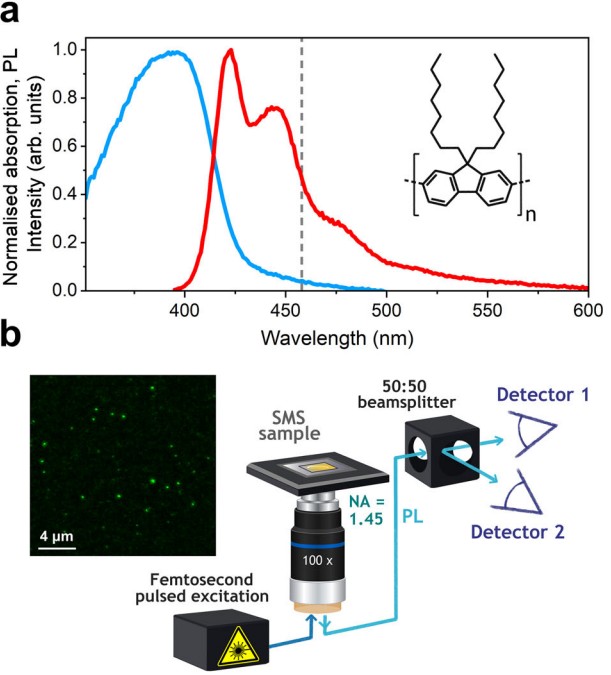

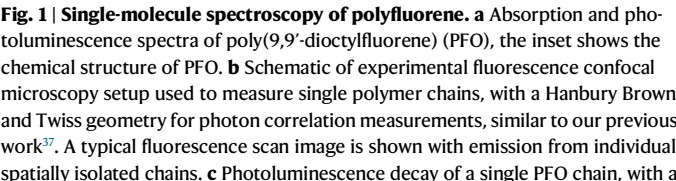

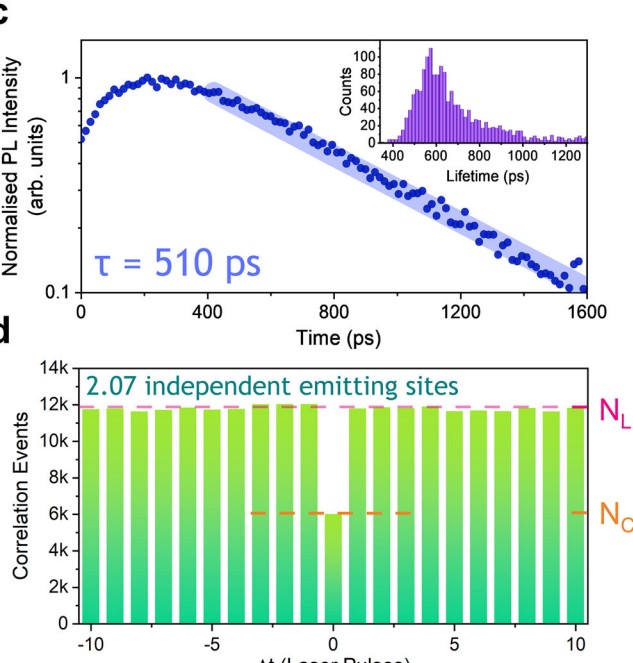

**Fig. 1 | Single-molecule spectroscopy of polyfluorene. a** Absorption and photoluminescence spectra of poly(9,9'-dioctylfluorene) (PFO), the inset shows the chemical structure of PFO. **b** Schematic of experimental fluorescence confocal microscopy setup used to measure single polymer chains, with a Hanbury Brown and Twiss geometry for photon correlation measurements, similar to our previous work[37]. A typical fluorescence scan image is shown with emission from individual spatially isolated chains. **c** Photoluminescence decay of a single PFO chain, with a fitted lifetime of 510 ps, shown with a semi-transparent thick straight line, while inset is a histogram of lifetimes for 1821 individual PFO chains. **d** Photon antibunching histogram obtained from 2215 individual PFO chains. The number of correlation events in the central bin at $\Delta t = 0$ is defined as $N_C$, and the mean number of correlation events in the lateral time-lagged bins as $N_L$. The ratio $N_C/N_L$ allows determination of the number of independent emitting sites, $n$, here calculated to be $2.07 \pm 0.09$.

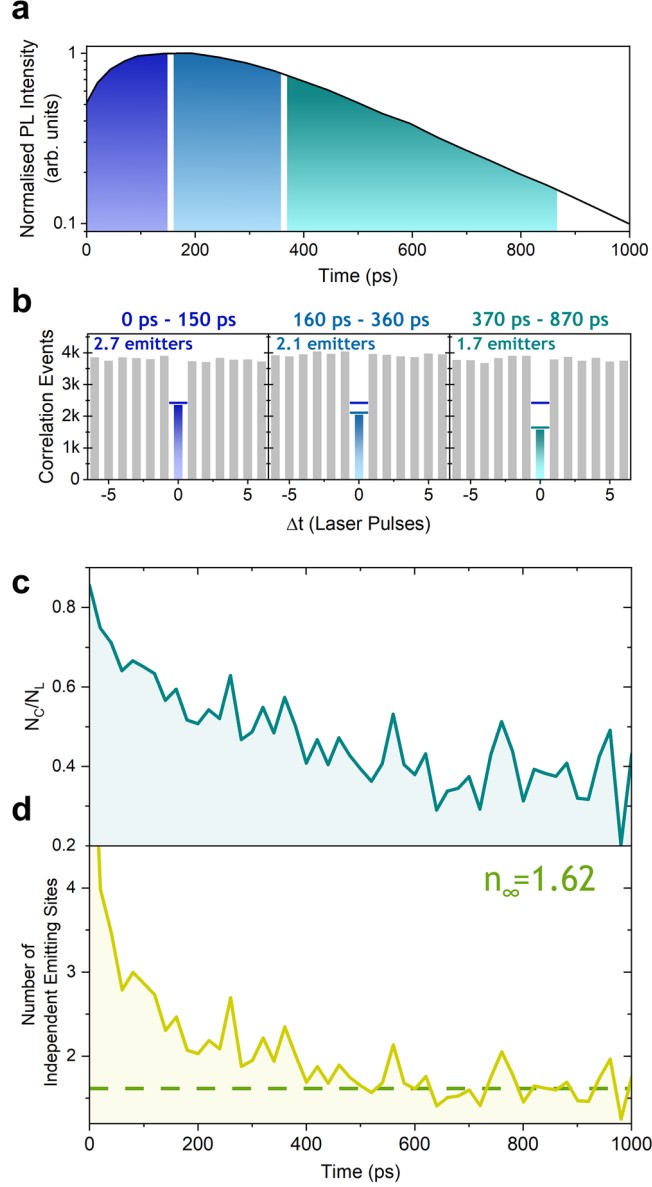

**Fig. 2 | Time-Resolved Photon Antibunching. a** The measured photo-luminescence decay summed over 2215 chains of PFO with the shaded regions indicating the photon arrival time windows used to construct corresponding antibunching histograms shown in (**b**). In the early window (0 to 150 ps), the dip in the central bin indicates 2.7 emitting sites, dropping to 2.1 in the 160 to 360 ps window, before dropping further to 1.7 emitters in the final window of 370 to 870 ps. The $N_C$ level for the first window is shown on all three plots to evidence the drop. **c** Plot of the continuously calculated $N_C/N_L$ ratio over the first 1 ns after the laser pulse with a resolution of 20 ps, showing the drop in the ratio. This is more comprehensible if the values are converted using Eq. 1 into the number of independent emitting sites, as shown in (**d**), with a drop from ~4 to ~1.6 over the first 600 ps.

more meaningful number of independent emitting sites via Eq. 1, which is plotted in Fig. 2d. Now, a direct readout of how many sites are active in the emission process in single isolated chains of PFO can be made across its excited state lifetime. What is found is that there is a reduction in this number, from ~4 to 1.6, over the first 600 ps, but then little further change out to at least 1 ns. This reduction directly tells us that excitons are interacting with each other on single chains. The most probable explanation for this is EEA. The timescale over which anni-hilation proceeds (0–600 ps) is suggestive that it is facilitated by exciton motion along the chain, and either two singlet excitons meet

and annihilate, or one moves and finds a relatively static second (low-energy trap site). Consequently, the results presented here offer an ability to observe on-chain exciton annihilation free from any back-ground population of non-annihilating excitons. To quantify this, we can compare the number of occurrences where two photons are detected after excitation ($N_C$) with the events where only one is. We find that this is 35 per million detected photons, i.e. the measured on-chain annihilation events in PFO are very rare and do not lead to any deviation in the conventionally observed photophysical properties (PL decay etc.), yet they do exist, and this measurement is uniquely sen-sitive to them.

Measuring single PFO chains provides a rare opportunity to extract valuable information on excited-state processes not readily accessible with ensemble techniques. In the results presented so far, we have neglected any consideration of heterogeneity in the chains, treating them as all behaving the same way. Many single-molecule spectroscopy studies indicate this is seldom the case for conjugated polymers[38–44]. To investigate this, we have combined a conventional Hanbury Brown and Twiss setup with a 3rd detector and split emission with a dichroic mirror centred at 458 nm, Fig. 3a. Ultimately, this introduces a means of measuring spectral fluctuations in a single PFO chain whilst simulta-neously allowing recording spectrally unbiased photon correlations. Now we can also monitor the spectral position of emission from PFO chains and selectively filter the TRAB correlation results for different circumstances. To allow assessment of how the spectral position of emission changes, we define a value of merit, the colour ratio:

$$Colour\ Ratio = \frac{I_{\lambda > 458} - I_{\lambda < 458}}{I_{\lambda > 458} + I_{\lambda < 458}} = \frac{C_1 - C_2}{C_1 + C_2} \quad (2)$$

where $C_1$ and $C_2$ are the recorded counts on the two detectors split by the dichroic mirror. Negative colour ratio values indicate emission dominated by $\lambda < 458$ nm, associated with ensemble steady-state PFO emission, while positive values indicate redder emission. Low-energy sites in PFO chains have been the subject of study over many years, with different explanations explored, including fluorenone keto-defects in PFO[45–48] or intrachain aggregates[49]. Plotting the colour ratio over 30 s for a typical single chain, Fig. 3b, we observe substantial variation. Here we have chosen a trace representative of the full range of values that a chain can take, but we see several different single-chain behaviours and show a number of them in Supplementary Information to allow better appreciation of the dynamical nature of the colour switching. The most common value is ~−0.2, consistent with −0.24 obtained by simulating transmission of a measured ensemble steady state PFO emission spectrum through the setup (see Supplementary Information for a full description of the simulation), which we will label as "blue sites". However, occasional and temporary jumps in the colour ratio are also observed, up to values close to 1. This red emission is consistent with fluorenone defects, with simulated expected colour ratio values for it emitting alone being 0.945 (spectrum from[48], see Supplementary Information for details), however, we note that such an assignment does not preclude the possibility of other explanations including intrachain aggregates on single chains[49] and so for simplicity we will label these as low-energy "red sites" rather than assigning them to specific species. The jumps to red emission are, however, fully reversible, indicating that conventional blue site PFO emission is recoverable, even after red sites have been detected on the chain. This can be explained when one considers that there are many possible absorption/emission sites on the chain, and the same ones may not necessarily be sampled on each subsequent laser pulse, and even if they are, our TRAB data indicates that there are multiple sites for emission to occur from. Thus, if absorption occurs close/next to a red site then immediate energy transfer to it will enable emission from it, but if absorption is farther away then this may not be possible or quick enough before blue emission has occurred. Additionally, there is the

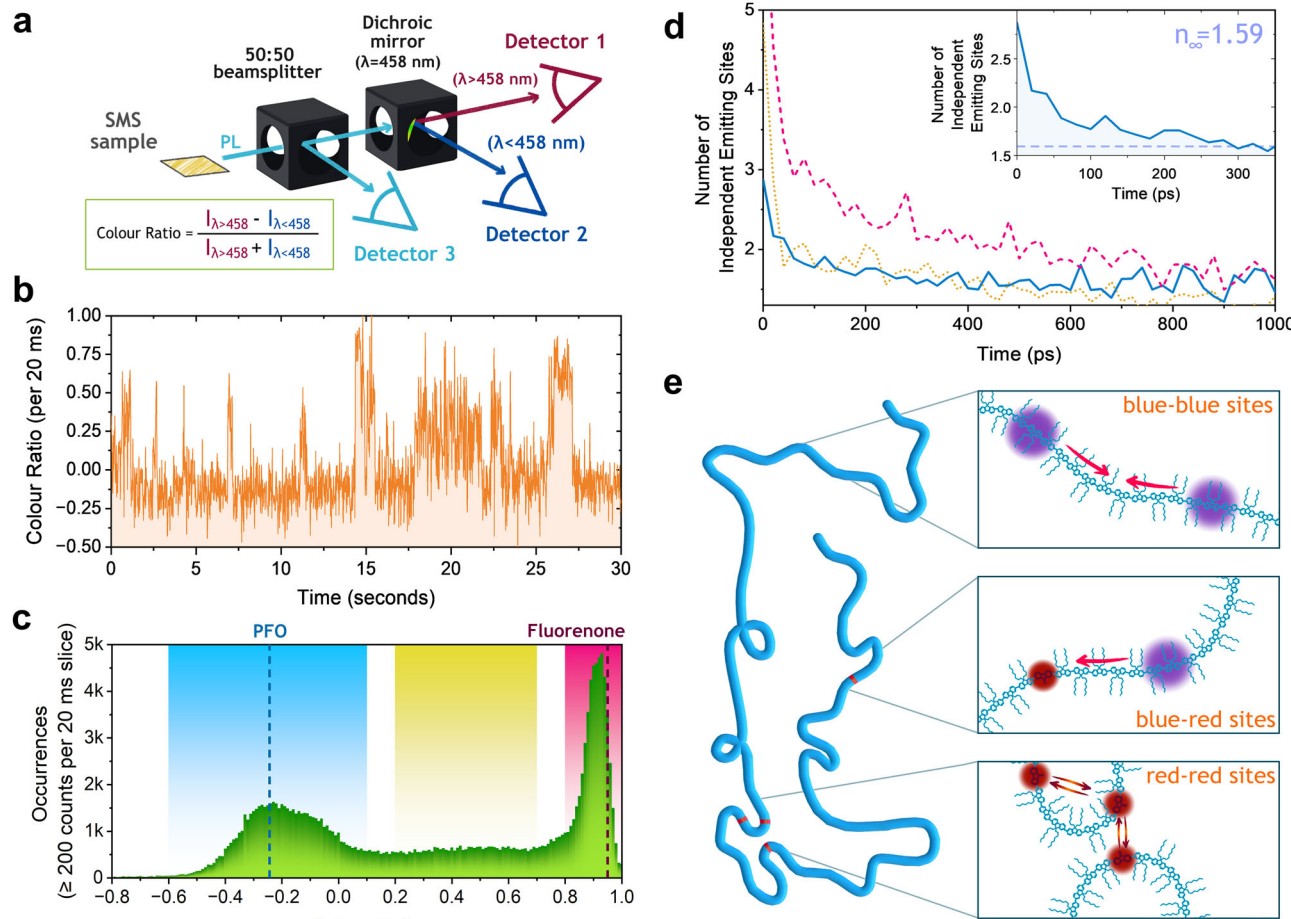

**Fig. 3 | Spectrally filtered time-resolved photon antibunching. a** Optical detection scheme enabling spectrally filtered Hanbury Brown and Twiss measurements. Detectors 1 and 2 can be used to define regions of time with a specific colour ratio (equation inset), derived from photons split on the dichroic mirror between them. Counts on these two detectors can then be summed and correlated with those on detector 3 for this region of time. **b** A single PFO chain time trace measured over 30 s, exhibiting significant fluctuations in colour ratio ranging from −0.25 to 1. **c** Histogram of colour ratio values per 20 ms bin with at least 200 counts present, for 5012 chains. Simulated colour ratios for steady state PFO (−0.24, blue

vertical dashed line) and fluorenone (0.945, pink vertical dashed line) emission spectra are also shown along with larger shaded regions. **d** Time-resolved photon antibunching derived plot of the number of independent emitting sites as a function of time for the three shaded spectral regions of colour ratio in panel c, with a zoomed-in view inset to aid clarity. **e** Schematic of a single polymer chain, with the three observed scenarios: blue site PFO excitons annihilating with each other (top), a blue site PFO annihilating with a low-energy red site (middle) and red sites annihilating with each other (bottom).

possibility of the formation of long-lived species that can shut down sections of the polymer chain from absorbing/emitting, thus enabling (or forcing) excitons to emit from other sites for some period of time. Typically, these dynamics are challenging to disentangle as the blocking species are non-emissive, so information about which excitonic sites are active is hard to obtain. However, here the red sites act as low-energy traps on the chain, and emission from them is spectrally separable from PFO[50], thus detection of their presence and effects can be used to determine exciton behaviour on single chains.

Sampling all measured values of colour ratio for 5012 chains with 20 ms binning with at least 200 counts present (see Supplementary Information for a full description of this conditional method) we can construct a histogram of the population, as shown in Fig. 3c. The expected values for emission solely from blue sites or red sites, as noted above, are shown as dashed lines, and two peaks in the histogram are observed, consistent with these two species. In general, energetic heterogeneity is expected when measuring single chains, so the broad distribution centred on −0.2 likely represents a variety of conformations of PFO. The red peak is narrow; however, this is most likely caused by the limited spectral discrimination due to the dichroic mirror's central wavelength being far away from this emission.

Existence of a population between the two peaks (shaded yellow) is not entirely unexpected and is because when we sample over 20 ms in this histogram region, we may have a mixture of blue site and red site emission present, giving an average value between the two. This explanation is favoured over the presence of any new third species, such as beta-phase segments[27], as they give a simulated colour ratio of 0.008 from their steady state PL spectrum, a value significantly lower than the range shaded yellow here.

As noted above, these spectrally-resolved single-chain measurements have been made with three detectors. Merging the recorded photons on detectors 1 and 2 in software post-measurement, we create a new virtual detector that can then be correlated with detector 3 for exploration of time-resolved photon antibunching with spectral sensitivity. Three highlighted regions from the colour ratio histogram have been chosen for this: blue sites (−0.6 to 0.1), red sites (0.8 to 1.0) and yellow for mixed (0.2 to 0.7). In software analysis routines (see Supplementary Information for a fuller description), we filter for periods of time when these colour ratios are present and then perform time-resolved photon antibunching, as previously described. The calculated number of independent emitting sites present versus time for the three highlighted regions are shown in Fig. 3d—we note that if no

spectral filtering is applied, TRAB results almost identical to those presented in Fig. 2 can be extracted (see Supplementary Information), and furthermore if we alter the conditions for the filtering (i.e. 10, 40 or 200 ms binning instead of 20 ms, and all counts per bin used instead of ≥200) no significant differences are observed, as shown in Supplementary Information. In considering these spectrally filtered time-resolved antibunching results, it is important to establish what we are measuring. When a specific spectral region is selected, then these measurements are sensitive to excited states in that energy window interacting with, and only with, each other, i.e. any interactions/losses with excitons of energies outside the filter window will not be registered. It is immediately clear that when chains have blue sites containing only PFO emission or mixed blue site, red site emission, they behave similarly, with a fast drop in the number of emitting sites, while chains that solely have red site emission show a slower decay. Describing each area in turn: when emission is from blue site PFO alone (Fig. 3d, blue data, solid line) they show fast reduction in the number of available emitting sites, in contrast to the spectrally unresolved results in Fig. 2. We are left with only two independent emitting sites present on the polymer chains (average $M_w ≥ 39,300$ g mol$^{-1}$, $M_n = 15,300$ g mol$^{-1}$, with an average degree of polymerisation of ~39 monomers per chain, see SI for the full molecular weight distribution) after only 100 ps. The time resolution here restricts full exploration of the early time behaviour, but at the earliest, we can resolve that there are only 3 or 4 emitting sites. A small, slower decay in the number of emitters (inset Fig. 3d) is observable and is consistent with on-chain EEA, but in general, very little annihilation takes place after 100 ps when blue site PFO chromophores alone are emitting. In contrast, for the periods of time when the chain is emitting from a red site (Fig. 3d, red data, dashed line), the number of independent emitting sites begins high (~5) and decays slowly, reaching only 2 after 1 ns. As noted above, in these spectral measurements, we are sensitive only to the interaction we filter for, i.e. here we are uniquely only going to observe how excited states situated on red sites interact with each other. In a similar way, the region intermediate between the two, (Fig. 3d, yellow data, dotted line), primarily represents encounter events between blue site PFO and low-energy red site excited states. This decay in the number of independent emitting sites follows the blue site PFO only closely, consistent with mobile PFO excitons defining the overall interaction. To confirm separability between the decay in the number of emitters for blue/yellow and red regions, we can calculate the error on each to provide ranges that they lie within, as shown in Supplementary Information.

We now turn our attention to the interpretation of these results, Fig. 3e. For conventional blue site PFO excitons interacting with each other, the loss of independent emitting sites is remarkably fast, close to complete in 100 ps along an entire conjugated polymer chain. The losses we see are real number losses, i.e. these represent EEA and a reduction in the ability for the single chain to host excitons and emit from multiple sites simultaneously. What we observe is the overall effective efficiency of annihilation on the chain; we note that while any individual encounter event between excitons may have a low probability of annihilation[51], if there are many encounter events, then the effective efficiency will still be high.

It is likely that the single chains we are measuring are not straight, so some pseudo-interchain coupling across different segments of the same chain will be present, enhancing exciton motion and annihilation. One way to explore this is by measuring with a range of defined chain lengths and comparing the annihilation behaviour. Unfortunately, such a study was beyond the scope of this work; however, we have explored this indirectly by using chain brightness as an analogue for length (see Supplementary Information for such extracted plots). We find that, as suggested immediately above, pseudo-interchain coupling is more likely in brighter (longer) chains, and thus they show more efficient annihilation for blue sites, while weaker (shorter) chains show less annihilation, as opportunities for excitons to hop to different chain segments are hindered. Control of the conformation of the chains would also be a good way to change the degree of interchain coupling, and this could potentially be achieved at the single-molecule level by changing the host matrix[52]. We conducted some initial early trials with a non-polar cyclic olefin host material; however, we found no obvious difference in basic photophysical properties, so did not pursue this further. If a host was found to strongly alter the PFO chain conformations and local environment (e.g. dielectric constant) then this would be a powerful way to explore interchain annihilation in further detail.

Our results have significant implications in defining limitations on light emission in PFO. Coupling between chromophoric sites is very strong at times <100 ps, leading to fast loss of population and ensuring a single chain is only able to support one or sometimes two excitons after this time. This puts an upper limit on how many photons can be emitted, whether through optical excitation or charge injection. In thinking about how to support higher exciton densities, structures that provide an energetic barrier surrounding an emitting centre to prevent motion/interaction between them are attractive. This can conveniently be tested with the low-energy red sites present in PFO, where we can exclusively look at how they interact with each other. Surprisingly, despite very likely only being a few[53] sites per chain spaced apart from each other, these sites interact, with a loss from 5 down to ~2 occurring, but over 1 nanosecond. As these are low-energy sites, it is anticipated that the barrier of surrounding chromophores with higher energy will ensure excitons are relatively trapped (from the PL peaks for both, this should be a barrier of ~0.5 eV). As noted above, we are not able to fully identify whether these sites are fluorenone keto-defects or intrachain aggregates, as some have suggested; however, the effect is the same either way. Consequently, we suggest that a combination of longer-range, slower FRET, and circumstances where two low-energy sites are close (either because they happen to be next to each other, or more likely across chain segments) explains the observed behaviour. This implies that attempts to create polymer chains with "protected" emitting sites using energy barriers will not easily work, as longer-range transfer and interchain pathways will generally ensure annihilation still occurs. Our results indicate that design strategies should focus on restricting the ability for interchain interactions that enhance annihilation pathways where high exciton densities are desired. However, our results also suggest that conjugated polymer chains show a remarkable degree of excitonic coupling, and this has positive implications for light-harvesting regimes such as solar energy conversion or light-matter interaction, where large macroscopic objects can behave as near-single objects.

## Discussion

In this work, we have observed very rare circumstances where EEA on a single conjugated polymer chain occurs for the first time. We find that these events are ~35 per million, i.e. we are measuring in a regime far away from high exciton densities, which are normally required to see annihilation. We find that excitons on chromophores consistent with conventional PFO emission interact with each other very readily. Three or four independent sites are present on the chain at the earliest times we can detect (~50 ps), and this reduces to ~2 in just 100 ps. When observing low-energy fluorenone defect or intrachain aggregate sites annihilating with each other, there is still a reduction down to ~2, but this takes ~1 ns. Consequently, these results suggest a fundamental limit in materials with a high degree of exciton coupling to support large exciton densities. Control of interchain coupling is identified as important if one wants to reach high exciton densities, particularly for chains in mesoscopic and bulk regimes found in devices.

## Methods
### Sample Preparation
Samples were measured on ~170 μm thick borosilicate glass slides, which were cleaned with a mixture of ultrapure water (Merck Direct-

Q3) and Hellmanex III to give 2% Hellmanex solution by flushing three times, followed by 20-minute sonication at 40 °C. These steps were repeated another two times using ultrapure water. $N_2$ was used to dry the slides prior to exposing them to UV ozone twice for 20-minute cycles in a UV ozone cleaner (Novascan, PSD-UV8). Single-molecule PL transients were recorded from individual polymer chains of poly(9,9'-dioctylfluorene) suspended (at approximately $1 \times 10^{-8}$ mg/ml) in inert thin film matrices of poly(methyl methacrylate) (PMMA). A 6% by weight 120 kDa PMMA:PFO:toluene mixture was deposited by spin-coating in an $N_2$ glovebox, giving 200 nm thick films. On the microscope, single chains were measured for 30 seconds each whilst being purged with a constant flow of nitrogen to prevent degradation.

## Single-molecule measurements

Single-molecule measurements were made on a home-built single-molecule microscope. Excitation is provided by the 80 MHz output at 800 nm from an ultrafast optical parametric oscillator (Coherent Discovery), which is routed through a pulse picker (APE PulseSelect) to give 40 MHz, with pulses of ~120 fs duration before being frequency doubled with a BBO crystal to give 400 nm. The beam is spatially expanded with a lens pair, then converted from linear to circular polarisation with a Berek Compensator (New Focus) set as a quarter waveplate for 400 nm. The beam is coupled into the microscope (Nikon Eclipse Ti2-U), reflected up to the sample with a dichroic (Thorlabs, DMLP425) before being focused with an objective (Nikon Plan Apochromat $\lambda$, 1.45 NA). Incident power on the sample is controlled with a neutral density wheel before the beam is expanded, with incident powers ~3 nW at the sample. The same objective collects emissions from single polymer chains. The sample is scanned with a 2D piezo stage (Physik Instrumente, P-733.2CD), with collected emission routed back through the dichroic, through a tube lens onto a pinhole (75 μm) to remove stray light. After recollimation, emission is then directed to the detection setup as described in the main text, with SPADs (MicroPhoton Devices, PD-100-CTE) in either a conventional Hanbury Brown and Twiss geometry with a 50:50 beamsplitter (Thorlabs BS013), or with an additional split between a 458 nm dichroic (Semrock, FF458-Di02) on one of the arms. Detected photons on the SPADs are recorded with a HydraHarp 400 (PicoQuant) on separate independent channels. Overall measurement control of the piezo and photon counting is made with home-coded software (C, Python and Qt framework). The data is stored as raw binary files in a first-in, first-out (FIFO) format, containing full time-tags (time with respect to the last laser pulse and time with respect to the start of measurement) for each photon. FIFO files are analysed in home-coded software (C, Python and Qt framework), enabling extraction/analysis of intensities, decays and photon statistics.

## Data availability

Datasets for each figure are available at https://doi.org/10.5525/gla.researchdata.2084 Source data are provided with this paper.

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

## Acknowledgements

G.J.H. acknowledges EPSRC (EP/V004921/1 and EP/V048805/1) and the Royal Society (RGS\R1\231392) for supporting this work. N.J.F.'s PhD was funded by an EPSRC DTP award (EP/T517896/1).

## Author contributions

N.J.F. co-devised the measurements, co-built the single-molecule setup, prepared the samples and made all single-molecule measurements. O.V. co-built the single-molecule setup and co-developed analysis methodologies. B.V.K.J.S. performed and analysed the GPC measurement. G.J.H. co-devised the measurements, co-built the single-molecule setup, coded the analysis methods and supervised the work. The manuscript was written by N.J.F. and G.J.H.

## Competing interests

The authors declare no competing interests.
