## [Transparent Peer Review file · Nature Communications]

Highly efficient exciton-exciton annihilation in single conjugated polymer chains

Corresponding Author: Dr Gordon Hedley

Version 0:

Reviewer comments:

Reviewer #1

(Remarks to the Author)

Fairbairn et al. report on exciton annihilation measured on single chains of the conjugated polymer polyfluorene (PFO). They find that despite the large molecular weight of the polymers the initial number of excitons reaches only 4, and this number decreases to 2 due to efficient annihilation process. The annihilation happens efficiently even when low-energy sites on the chain are populated.

These findings are novel and important for the conjugated polymer community, and the experiments are seemingly done carefully on a high technical level. I would be supportive of publication, provided the authors address the points below:

1. The authors claim high-efficiency annihilation, but can they actually estimate the efficiency? There has been a recent report on a different conjugated polymer which shows that when two excitons meet on the chain the probability of the annihilation is only 10 % (Nature 2023, 616, 280; <https://doi.org/10.1038/s41586-023-05846-7>)
2. When discussing the nature of the low-energy sites the authors automatically assume a keto defect. I do not mean to start discussion on the origin of the green emission here, but it would be fair to mention other possibilities. It has been directly demonstrated recently by combined afm and single-molecule fluorescence that intrachain aggregate is responsible for at least some of the observed low-energy sites (ACS Nano 2023, 17, 8074; <https://doi.org/10.1021/acsnano.2c09773>)
3. Assuming the intrachain aggregate origin of the low-energy sites would actually make it much easier to explain some of the dynamic phenomena the authors observe. The authors are finding that the emission switches reversibly between the high and low-energy sites on timescales of seconds. To explain this reversible nature using the keto defect model they have to assume that the energy does not reach the defect (which is present at all times) because of a kind of unspecified 'long-lived species that can shut down sections of the polymer chain from absorbing/emitting'. It is not clear what would be the origin of such reversible species, but to completely shut down the energy transfer to the defect, the defect has to be surrounded by the 'long-lived species', and these have to extend beyond the Forster radius. Given that this radius is on the order of 5 nm in PFO, a single keto defect would block at least a 10 nm sphere within the chain from absorbing/emitting light. Further, it has been shown (JPCB 2008, 112, 12575; <https://doi.org/10.1021/jp806963u>) that conjugated polymers of similar molecular weight in poor solvent (and PMMA is poor solvent for PFO) possess sizes (in terms of radius of gyration) between 20 and 30 nm, which means that two keto defects on the chain would practically disable the whole chain from interacting with light. But purely blue emission (high-energy site) is observed for most of the time (Fig. 3b), which means that for most of the time these 'long-lived species' have to be active, and effectively decrease the fraction of the chain that interacts with light to tens of % (?). The alternative explanation that the low-energy site is being repeatedly formed and dissociated, as a conformation-change-induced intrachain aggregate, is a much simpler and natural explanation. Without the site, the whole chain emits the blue fluorescence, when the aggregate is formed the energy flows to the low-energy site, and the flow stops when the aggregate disintegrates.
4. Other points: the initial number of excitons should depend on the excitation intensity, have you measured the intensity dependence? Also, the PDI of the PFO seems to be high (~ 4), how does this fact affect the results? More information on the PFO itself should also be provided in the SI.

Reviewer #2

(Remarks to the Author)

This work employs the picosecond time-resolved antibunching (TRAB) technique previously developed and well characterised by the authors (Hedley, G. J. et al., Nat. Commun. 12, 1327 (2021)) to study FRET and exciton-exciton

annihilation (EEA) processes in single, isolated conjugated polymer chains. EEA is a fundamental loss mechanism in optoelectronic devices with high excitation densities. For example, EEA prevents low threshold lasing in organic semiconductors and reduces the operational efficiency of high-brightness OLEDs. This work is important because it offers a new technique to study EEA: the use of TRAB, especially with the incorporation of multiple detectors and a dichoric filter to study transfer onto defects, offers a powerful assay in the study of EEA in isolated conjugated polymers. In general, measurements of EEA are prone to artefacts that require accounting for and are typically studied using samples where intermolecular EEA dominates. The technique here appears to be a more direct measurement of EEA with fewer artefacts. They use this technique to characterise intramolecular EEA on single conjugated polymer chains, as opposed to the much better studied intermolecular processes. The findings suggest that it is difficult to avoid EEA completely and that strategies to overcome EEA, such as isolation of conjugated polymers (e.g. Beilstein J. Org. Chem. 10, 2145–2156 (2014)), are not sufficient.

The results are of high quality, and a real strength of the work is the use of statistics, enabled by studying thousands of single molecules. This has allowed the authors to determine the specific impact of intramolecular EEA: by measuring the photon statistics, they are able to demonstrate that the roughly 4 initial excitons that reside on a single molecule rapidly self-annihilate, leaving one or two excitons. The data and analysis in the paper seems mostly robust. However, it is unclear to me in some cases (e.g. Figure 3) whether (or how) selection of the data to do photon statistics alters these conclusions. For example, what happens if the authors use a larger binning (e.g. 50ms), or stricter (or less strict) photon counts threshold for rejection, or more finely-grained spectral regions?

The polydispersity of the PFO used also seems quite large ($M_w > 200$ kDa, $M_n = 54$ kDa). How does this affect the results? For example, could the results be skewed by the fact that polymers are all different lengths and that the brighter polymers (i.e. most likely to be measured) are also likely to be the longest? What is the shortest PFO chain that can support EEA? Would using a low polydispersity sample where all molecules are a fixed length, for example, allow you to get information about single molecular exciton diffusion (lengths, diffusion coefficient, etc.)?

In addition, the paper casts a pessimistic outlook for the use of conjugated polymers in applications requiring high excitation densities, suggesting that intramolecular EEA places a 'fundamental limit' on maximal exciton densities in such systems. Do the authors have any suggestions for mitigation strategies? Alternatively, can we exploit such annihilation processes to our benefit in some way?

The writing and presentation of the paper is commendably clear and overall, the work is of sufficient quality and interest for publication in Nature Communications, although I list below a few suggestions to improve it further that should be completed before publication.

1. Although it may seem obvious, I suggest the authors highlight in the abstract that this study is on single, isolated conjugated polymers, and so studying intramolecular EEA, which is a novel aspect of this work.
2. Is the time resolution (~50ps) estimated, from the maximum SPAD (MicroPhoton Devices, PD-100-CTE) NIM timing resolution (apparently 50ps), or measured? Could an IRF trace for the TCSPC be shown in the Supplementary Information? Presumably this is the constraint for the IRF/time resolution on the TRAB; is this correct? Is it possible to deconvolve the TRAB traces as some do with TCSPC traces?
3. Figure 1c – is the shaded thick light-blue line indicating something? Also, this trace looks like it has a different 'zero time' to Figure 2a; is there a reason for this?
4. Figure 1 caption – might be good to insert a reference to the previous work (Hedley, G. J. et al., Nat. Commun. 12, 1327 (2021)) in the description of panel b.
5. Figure 2a – I'm not sure if the TCSPC kinetic trace here is an average over thousands of chains (contrasting with Figure 1c, which is a single chain). Can this be briefly clarified in the figure caption (or the figure itself)?
6. Figure 2b,c,d – what is the histogram binning time here? 20ms?
7. Figure 3e – colours used for the 'PFO' vs. 'keto' excitons may be difficult for people with colour blindness (specifically, deuteranopia and protanopia) to distinguish. I suggest revising.
8. Throughout – I notice 'polyfluorene' is sometimes used, in addition to 'PFO'. Is a distinction intended?
9. Discussion – 'We find that these events are ~35 per million, i.e. we are measuring in a regime far away from high exciton densities which are normally required to see annihilation.' The generation of 4–5 excitons per polymer per laser pulse sounds like a high exciton density. Can the authors provide or calculate the events ratio in a high exciton density regime, to give assurance their work is indeed a low (or intermediate) density?
10. Supplementary Information – do any of the optics (e.g. the dichoric mirror) have a non-negligible absorption component (in addition to the considered transmitted and reflected components)? Is it safe to disregard this absorption?
11. Am I correct in saying the 20ms binning corresponds to 8×10^5 laser pulses (i.e. has correspondence to ' Δt ')? It would be a useful clarification if these two times can be linked (given we are already having to consider TCSPC microtime also).

Reviewer #3

(Remarks to the Author)

The work by Fairbairn and coworkers presented an interesting study of the excitonic dynamics of conjugated polymers. The time-resolved photon correlation experiment revealed rapid and efficient exciton-exciton annihilation at the single polymer molecule level, indicating that achieving high exciton density in conjugated polymers is difficult. Overall, this work is of high quality and can be accepted and published in Nature Communications. However, some discussions regarding the following aspects should be discussed/addressed before acceptance for publication.

1. The authors studied single PFO chains with a molecular weight of $M_w \sim 200$ kDa, which most likely supports both interchain and intrachain interactions. It would be interesting to differentiate interchain and intrachain excitons via studying a relatively lower M_w single polymer chain (e.g., tens of kDa or lower) and/or a single aggregate of low- M_w chains. A relatively clearer picture of exciton interaction physics can be obtained, which could offer some potential strategies in materials or device perspectives to achieve high exciton densities for brighter polymer-based OLEDs or lasers.
2. Did the authors check the polymer matrix (e.g., dielectric) effect on single PFO chain behavior? The matrix environment strongly affects the EEA in terms of exciton diffusion, exciton binding energy, and energy transfer processes. Some discussion or investigation on this aspect should be added.
3. What is the noise level of time-gated antibunching data? Was the noise corrected for the presented data?

Reviewer #4

(Remarks to the Author)

Version 1:

Reviewer comments:

Reviewer #1

(Remarks to the Author)

I thank the authors for a comprehensive revision of the work which addresses all my points, and want to congratulate them for these excellent results.

Reviewer #2

(Remarks to the Author)

The authors have done a good job of answering the comments and in my view the manuscript is now suitable for publication in Nature Communications.

Reviewer #4

(Remarks to the Author)

We thank the reviewers for their supportive, detailed and insightful comments, and are pleased that they see the importance and value in our work. In this submission we have provided a revised manuscript, both with and without track changes visible, and for convenience for each of the changes requested by the reviewer we have summarised them all here in this document. Responding to the specific queries/suggestions from each reviewer in-turn:

Reviewer 1:

“1. The authors claim high-efficiency annihilation, but can they actually estimate the efficiency? There has been a recent report on different conjugated polymer which shows that when two excitons meet on the chain the probability of the annihilation is only 10 % (Nature 2023, 616, 280; <https://doi.org/10.1038/s41586-023-05846-7>)”

This is a good point that the reviewer makes. There is a subtle distinction with the word efficiency that is important to describe. It can mean the chance of annihilation in a single encounter event irrespective of how often that encounter happens, or the overall chance of two excitons annihilating irrespective of how often or little they encounter each other. We are measuring the latter definition in our work, i.e. we do not know how often the excitons encounter each other, but we observe the overall efficiency of annihilation (perhaps this could be described as the effective efficiency) because we are sensitive to the probability of two excitons being on the chain. We have added the following text to the manuscript (P15), referencing the highlighted paper, to capture this distinction:

“What we observe is the overall effective efficiency of annihilation on the chain; we note that while any individual encounter event between excitons may have a low probability of annihilation,⁵¹ if there are many encounter events then the effective efficiency will still be high.”

“2. When discussing the nature of the low-energy sites the authors automatically assume a keto defect. I do not mean to start discussion on the origin of the green emission here, but it would be fair to mention other possibilities. It has been directly demonstrated recently by combined afm and single-molecule fluorescence that intrachain aggregate is responsible for at least some of the observed low-energy sites (ACS Nano 2023, 17, 8074; <https://doi.org/10.1021/acsnano.2c09773>)”

We appreciate the comment from the reviewer and acknowledge that this topic has been the subject of many studies over the years, often with convincing yet contradictory conclusions found when comparing between them. Likewise, we were aiming not to start a discussion on the source of the low-energy emission in PFO, but rather to explore how even when such low energy sites are present, annihilation still

dominates between them in single chains. We are happy to ensure the readers are aware that there are multiple explanations for the presence of the low-energy sites, we have modified the text to include other possibilities (P9):

“Low-energy sites in PFO chains have been the subject of study over many years, with different explanations explored, including fluorenone keto-defects in PFO⁴⁵⁻⁴⁸ or intrachain aggregates.⁴⁹”

Additionally, to avoid confusion we have altered our nomenclature in the text to refer to “blue sites” (meaning equivalent of steady state PFO emission), “red sites” meaning low-energy emission (from either keto-defects or intrachain aggregates) and “yellow sites” meaning a mixture of blue and red sites. We have updated the text throughout to follow this convention which avoids restricting discussion to only one possible source of red emission. There are many changes throughout the manuscript as a result of this and we won’t list them all here, but they are best summarised in the text where the red sites are first discussed (P10):

“This red emission is consistent with ~~emission by~~ fluorenone defects, with simulated expected colour ratio values for it emitting alone being 0.945 (spectrum from 48, see Supplementary Information for details), however, we note that such an assignment doesn’t preclude the possibility of other explanations including intrachain aggregates on single chains⁴⁹ and so for simplicity we will label these as low-energy “red sites” rather than assigning them to specific species.”

Additionally, we have included the possibility of intrachain aggregates in our discussion to indicate that neither possibility is specifically excluded (P16):

“As noted above, we are not able to fully identify whether these sites are fluorenone keto-defects or intrachain aggregates as some have suggested, however, the effect is the same either way. Consequently, we suggest that a combination of longer-range slower FRET, and circumstances where two ~~fluorenone~~ low-energy sites are close”

Parsing and paraphrasing query 3, the reviewer suggests that the proposed switching of emission in single chains is not consistent with keto-defects as this would require blocking species that would eliminate the entire chain from emitting, and that these blocking species would need to be present most of the time in order to primarily have blue emission. They instead seem to suggest that dynamical intrachain aggregate sites are responsible e.g. they state: “*Without the site, the whole chain emits the blue fluorescence, when the aggregate is formed the energy flows to the low-energy site, and the flow stops when the aggregate disintegrates.*”

We thank the reviewer for these suggestions. Firstly, it is important to note that our results indicate that the entire chain is not behaving solely as a single object, especially at early times in the excited state lifetime. We know that there are 3-5 independent chromophoric sites on the chain simultaneously, and furthermore we have no idea if those are the same sites on each laser pulse or continually vary. We have work in progress on another polymer using polarisation measurements that indicate the sites that emit photons are constantly different, i.e. we are continually sampling different sites on the chain (and our photon statistics tells us here that we have the choice of 3-5 sites every time anyway). So for blue PFO emission to be detected by us here it is not true that all lower-energy sites on the chain would have to be suppressed/blocked for extended periods of time, because there are multiple independent sites on the chain (and with photon statistics we know these really are independent of each other, for if they were not then they would behave as one and we would have perfect antibunching at all times).

The query thus comes down to rationalising why the observed colour ratio (emission wavelength) varies. In the main text we alluded to blocking species that shut down portions of the chain for periods of time, which the reviewer thinks are unfeasible for extended durations. In Figure 3 we show only a single time trace. We chose one that enabled close to the full range of colour ratio values to be evidenced; however, all manner of dynamical behaviour exists across the ~5000 that were measured; sometimes the ratio starts on red sites and switches to blue ones, sometimes the other way around, sometimes it is stable on only one colour for extended periods, and sometimes it is constantly changing. We have added some examples of these to SI and shown below to better evidence the variability in this dynamical behaviour, and added the following to the main text (P9):

“Here we have chosen a trace representative of the full range of values that a chain can take, but we see several different single-chain behaviours and show a number of them in Supplementary Information to allow better appreciation of the dynamical nature of the colour switching.”

We suggested the presence of blocking species as controlling where the emitting sites are from, however the reviewer is correct that we cannot know this with certainty, and it is potentially complex and additionally may be related to chain conformation etc. Thus we have updated the text to provide the simpler explanation as discussed above, i.e. we have multiple absorbing and emitting sites on the chain, and thus randomly which ones are active is entirely dependent on stochastic behaviour, at most possibly only aided by blocking species, but not entirely controlled by them. We know that energy transfer to a single low-energy site on the entire chain is not dominant, for if it was then no other emission would be observed, and we would observe perfect antibunching. We have thus updated the text to reflect this (P10):

“The jumps to red emission are, however, fully reversible, indicating that conventional blue site PFO emission is recoverable, even after red sites have been detected on the

chain. This can be explained when one considers that there are many possible absorption/emission sites on the chain and the same ones may not necessarily be sampled on each subsequent laser pulse, and even if they are, our TRAB data indicates that there are multiple sites for emission to occur from. Thus, if absorption occurs close/next to a red site then immediate energy transfer to it will enable emission from it, but if absorption is farther away then this may not be possible or quick enough before blue emission has occurred. Additionally, there is the possibility of the formation of long-lived species that can shut down sections of the polymer chain from absorbing/emitting, thus enabling (or forcing) excitons to emit from other sites for some period of time.”

Finally, we are a little confused with the suggested mechanism by the reviewer involving aggregates forming/disintegrating. Our polymer chains are measured in the solid state in a PMMA matrix. While some small amounts of chain motion will occur, the polymer chain is not free to move as if it were in solution. Thus, we find it hard to rationalise the suggestion that polymer chains are continually moving to form and disintegrate aggregates in the solid state. We have no other evidence to support this in our measurements and apply Occom’s razor here.

“4. Other points: the initial number of excitons should depend on the excitation intensity, have you measured the intensity dependence? Also, the PDI of the PFO seems to be high (~ 4), how does this fact affect the results? More information on the PFO itself should also be provide in the SI.”

Some intensity dependence measurements were completed as part of initial studies on this polymer but not using the full techniques we have presented in this work. However, we note that our measurement is very largely insensitive to this. A fuller explanation is given in response to a query by reviewer 2, but in brief, our antibunching method tells us the number of emitting sites present, not the number of excitons. While we see 3-5 independent emitting sites at time zero, we do not have 3-5 excitons on the chain, indeed we almost always only have 1 at most. Varying the excitation density will lead to more instances of two excitons on the chain, but with our current excitation density this number is 35 per million, so even a factor of 10 increase in density will not lead to any significant difference in instances of observed annihilation (and even then the annihilation rate should not change). Working at higher excitation density will simply increase the chance of multiphoton absorption or other processes that increase the risk of photodegradation.

On the PDI and possible effects of the molecular weight on the photophysics, we have conducted a brightness analysis of the recorded data, this is discussed in more detail below in response to reviewer 3. We have additionally performed GPC on the polymer

to obtain the full molecular weight distribution; this has been added to SI and is shown below. We have added Bernhard Schmidt as a co-author as he performed these measurements. Interestingly, we find the original numbers quoted, obtained from Sigma Aldrich upon purchase of this material, do not match with what we have measured here. There is one dominant peak in the distribution, representing a M_n of 15100 g mol^{-1} and M_w of 30400 g mol^{-1} and a dispersity (\mathcal{D}) of 2.01, and a much smaller high molecular weight peak, with M_n of $471000 \text{ g mol}^{-1}$ and M_w of $555000 \text{ g mol}^{-1}$ and a dispersity (\mathcal{D}) of 1.18. The overall distribution, accounting for both thus gives M_n of 15300 g mol^{-1} and M_w of 39300 g mol^{-1} , and a dispersity (\mathcal{D}) of 2.56, and these are the values we have quoted in the updated main text (P12).

Reviewer 2:

We thank the reviewer for their succinct summary of why our work is important and agree that studying intrachain exciton-exciton annihilation free from artefacts can aid

understanding of fundamental behaviour in conjugated polymers. This is one of the primary motivations for our study. Turning to the queries the reviewer has raised:

“However, it is unclear to me in some cases (e.g. Figure 3) whether (or how) selection of the data to do photon statistics alters these conclusions. For example, what happens if the authors use a larger binning (e.g. 50ms), or stricter (or less strict) photon counts threshold for rejection, or more finely-grained spectral regions?”

There is of course no absolute way to analyse the data free from selection bias. If we take a larger temporal binning then we start to reduce our sensitivity to spectral information, which ultimately leads back to the data observed in Figure 2 if we take very large bins (e.g. 5 or 10 seconds). Conversely if we bin on shorter time windows then we are more susceptible to noise and thus spectral information is also lost.

To test this, we have taken the data from Figure 3d and analysed it with 10, 40, and 200 ms binning (along with the original 20 ms binning) and little differences are observed. We have scaled the photon count threshold for each different time window to ensure that we are not selecting different intensity ranges, i.e. we have used ≥ 100 , ≥ 400 and ≥ 2000 counts per window respectively to match with the ≥ 200 counts used in the original 20 ms binning. We have added this plot to SI and commented on this in the main text (P12) to evidence the insensitivity:

“filtering is applied, TRAB results almost identical to those presented in Figure 2 can be extracted (see Supplementary Information), and furthermore if we alter the conditions for the filtering (i.e. 10, 40 or 200 ms binning instead of 20 ms, and all counts per bin used instead of ≥ 200) no significant differences are observed, as shown in Supplementary Information. In considering...”

As noted in the SI, the photon count thresholds are required for the colour ratio histogram because otherwise we have different overlaid distributions caused by what we internally refer to as “integer division” problems, e.g. if there are only 10 photons in a 20 ms slice then the colour ratio can only take 11 values across the full range of -1 to +1 (10-0, 9-1, ... 5-5, ... 1-9, 0-10). Thus, when taking many different time slices with many different numbers of photons present in the slice we end up with prominent artefact peaks caused by slices with specific numbers of fewer photons. We anticipate that the count threshold will be relatively insensitive if we reduce it because photon correlations scale as intensity squared, so on the lower side little difference should be observed. To test this, we have run the analysis with no minimum threshold (i.e. all photons are used) and see little difference. We have updated the SI to show this plot, and it is shown below for convenience.

We note that the original lower limit of 200 counts per 20 ms is effectively a gentle filter to remove the lowest intensity data. In response to the reviewer’s next comment and

those by reviewer 3, we have conducted a full intensity filtering (i.e. with upper limits as well) and do see some differences, which is somewhat related to this point, please see our response to reviewer 3 for this information. This includes a plot of an intensity histogram indicating that the lower-limit filter we use in Figure 3d (≥ 200 counts per 20 ms, which is equivalent to 10 kHz) only removes the lowest intensity traces from the analysis.

“The polydispersity of the PFO used also seems quite large ($M_w > 200$ kDa, $M_n = 54$ kDa). How does this affect the results? For example, could the results be skewed by the fact that polymers are all different lengths and that the brighter polymers (i.e. most likely to be measured) are also likely to be the longest? What is the shortest PFO chain that can support EEA? Would using a low polydispersity sample where all molecules are a fixed length, for example, allow you to get information about single molecular exciton diffusion (lengths, diffusion coefficient, etc.)?”

The reviewer raises a valuable point, and we have covered this in response to reviewer 3, below, by performing brightness-dependent analysis as a substitute for fixed chain-length measurements. The outcomes of that analysis hint that longer chains may support more interchain diffusion and thus enhance EEA between blue sites. Consequently, determining specific diffusion coefficients etc. may be possible in the future, but with the caveat that it could well be very chain-length and conformation dependent.

“In addition, the paper casts a pessimistic outlook for the use of conjugated polymers in applications requiring high excitation densities, suggesting that intramolecular EEA places a ‘fundamental limit’ on maximal exciton densities in such systems. Do the authors have any suggestions for mitigation strategies? Alternatively, can we exploit such annihilation processes to our benefit in some way?”

We appreciate the sentiment expressed and agree. Mitigation strategies are likely to involve good understanding and design of the energetic and conformational landscape in polymers; indeed, we are currently writing a separate paper on a different polymer

that directly observes this and does give hope that higher densities can be sustained in specific ways. Our new brightness analysis hints that if interchain coupling can be controlled/supressed then this does offer hope for enabling high densities.

Exploitation of the efficient annihilation can be possible, but we did not make clear linkages to such scenarios in the original manuscript. The efficient annihilation essentially tells us that significant parts of polymer chains are efficiently coupled together, this has benefits in domains where light emission is not required, most clearly in light harvesting. If large parts of the chain are coupled and energy funnelling to low-energy sites occurs then this is consistent with the surprisingly good performance of conjugated polymers in organic photovoltaic cells (we say surprising, because many other properties of these materials, e.g. low dielectric constants and charge mobilities, often work against high performance). We have updated the main text (P16-17) to better contextualise a roadmap for improvements and highlight positive outcomes for this behaviour:

“This implies that attempts to create polymer chains with “protected” emitting sites using energy barriers will not easily work, as longer-range transfer and interchain pathways will generally ensure annihilation still occurs. **Our results indicate design strategies should focus on restricting the ability for interchain interactions that enhance annihilation pathways where high exciton densities are desired. However, our results also suggest that conjugated polymer chains show a remarkable degree of excitonic coupling, and this has positive implications for light-harvesting regimes such as solar energy conversion or light-matter interaction, where large macroscopic objects can behave as near-single objects.**”

And in the Discussion at the end (P17):

“Consequently, these results suggest a fundamental limit in materials with a high degree of exciton coupling to support large exciton densities. **Control of interchain coupling is identified as important if one wants to reach high exciton densities, particularly** ~~when our observations are the best case scenario, free from the very significant interchain interactions that would be present~~ **for chains** in mesoscopic and bulk regimes found in devices.”

“1. Although it may seem obvious, I suggest the authors highlight in the abstract that this study is on single, isolated conjugated polymers, and so studying intramolecular EEA, which is a novel aspect of this work.”

We are happy to do this and have updated the abstract:

“Here we have measured the time-resolved photon statistics of single, isolated chains of polyfluorene to extract the absolute number of independent emitting sites present and its time dependence by studying the intramolecular exciton-exciton annihilation. We find that after...”

“2. Is the time resolution (~50ps) estimated, from the maximum SPAD (MicroPhoton Devices, PD-100-CTE) NIM timing resolution (apparently 50ps), or measured? Could an IRF trace for the TCSPC be shown in the Supplementary Information? Presumably this is the constraint for the IRF/time resolution on the TRAB; is this correct? Is it possible to deconvolve the TRAB traces as some do with TCSPC traces?”

The nominal time resolution is from the SPAD datasheet. Our measured values differ a little as we are typically observing a little further in the blue where the transit time spread is slightly longer and has a slower artefact. The reviewer is correct that the detector response time is the constraint for our overall time resolution on the TRAB. We have added IRFs to SI to show the time resolution we are working with, and they are also shown below for convenience. Deconvolution of TRAB traces is a little tricky and not yet something we have formulated full methodologies for. As the TRAB trace ultimately comes from a division of two decays (those on the central and lateral correlation peaks) we would need to use an autoreconvolution method (equivalent to how some TCSPC anisotropy data is handled). An additional complication is the subtle spectral dependence in the SPAD IRFs, which would also need to be accounted for. In short, deconvolution is this is not yet possible, but we aim to explore this in the future and is the reason we do not fit specific time constants to the data.

“3. Figure 1c – is the shaded thick light-blue line indicating something? Also, this trace looks like it has a different ‘zero time’ to Figure 2a; is there a reason for this?”

We apologise for any confusion, the thick light-blue line is the fit to the decay, with the tau value for this fit shown bottom left of the panel. We have made this clearer in the figure caption to avoid any misunderstanding. As the reviewer will know, time-zero is just convention when deconvolution fitting is not used; for other data we have followed one that sets this to be half-way up the rise-time of the normalised PL (i.e. when PL intensity = 0.5). The plot in Figure 1c was not fully following this convention and we have now updated it to do so, we thank the reviewer for making us aware of the error.

“4. Figure 1 caption – might be good to insert a reference to the previous work (Hedley, G. J. et al., Nat. Commun. 12, 1327 (2021)) in the description of panel b.”

We are happy to do this and have inserted the reference.

“5. Figure 2a – I’m not sure if the TCSPC kinetic trace here is an average over thousands of chains (contrasting with Figure 1c, which is a single chain). Can this be briefly clarified in the figure caption (or the figure itself)?”

We are happy to do this, the decay represents a sum of the decays of 2215 chains, used to show the antibunching regions used. We have updated the figure caption to clarify this.

“6. Figure 2b,c,d – what is the histogram binning time here? 20ms?”

If the enquiry is what the correlation histogram binning is then we are happy to clarify that this is laser pulse periods, so 25 ns per histogram bin. If the reviewer is enquiring what the intensity binning is, then for Figure 2 there is no histogram data used to bin, the traces are simply taken as measured and correlations calculated across the full measurement window for each chain, free from any information on macroscopic (ms-s) times.

“7. Figure 3e – colours used for the ‘PFO’ vs. ‘keto’ excitons may be difficult for people with colour blindness (specifically, deuteranopia and protanopia) to distinguish. I suggest revising.”

We thank the reviewer for bringing this to our attention and have updated Figure 3e to correct this. Furthermore, we will keep these comments in mind for future paper figures as we had not considered it as much as we should have with the routine usage of colour in electronic papers.

“8. Throughout – I notice ‘polyfluorene’ is sometimes used, in addition to ‘PFO’. Is a distinction intended?”

There was no distinction implied when referring to either ‘PFO’ or ‘polyfluorene’. That is, these are used interchangeably, however, we understand that this might create ambiguity when read and so we are happy to use ‘PFO’ throughout the text for consistency.

“9. Discussion – ‘We find that these events are ~35 per million, i.e. we are measuring in a regime far away from high exciton densities which are normally required to see annihilation.’ The generation of 4–5 excitons per polymer per laser pulse sounds like a high exciton density. Can the authors provide or calculate the events ratio in a high exciton density regime, to give assurance their work is indeed a low (or intermediate) density?”

One of the primary benefits of using photon correlations for exploring emitter number science is that we can avoid the densities normally required in conventional population-based measurements (e.g. transient absorption or emission). In our measurements we never have 4 or 5 excitons on the chain at once, indeed we almost never have even 2 (we have 2 excitons on the chain at a rate of 35 per million instances of 1 exciton on the chain). What we do instead is use that tiny chance of two per chain to tell us how many sites the chain can support. Throughout the manuscript we have been careful to use the phrase “independent emitting sites” when describing what we are measuring with TRAB. This is very deliberate, as we are not directly detecting 3,4 or 5 excitons on the chain, but rather we are measuring the chance of detecting two emitted photons (from two excitons), which tells us how many the chain *could* support at once. In qualitative terms, as the number of independent sites increases, the chance of detecting two photons increases, and we can use that to determine absolute numbers. We hope this makes sense, it is a confusion we have occasionally had in internal discussions ourselves in the past and are now careful to avoid.

“10. Supplementary Information – do any of the optics (e.g. the dichoric mirror) have a non-negligible absorption component (in addition to the considered transmitted and reflected components)? Is it safe to disregard this absorption?”

To the best of our knowledge there should be no significant absorption from the optical components used in the setup. Even if this did occur, the transmission spectrum takes this into account in the ways that matter for the colour ratio calculation we do and the only concern would be any subsequent emission (from the optical element that has absorbed), which is deemed very unlikely.

“11. Am I correct in saying the 20ms binning corresponds to 8×10^5 laser pulses (i.e. has correspondence to Δt)? It would be a useful clarification if these two times can be linked (given we are already having to consider TCSPC microtime also).”

The reviewer is correct, a 20 ms bin gives 8×10^5 laser pulses. Our Δt is values of laser pulse periods, and we only really use a few (e.g. in Figure 2b we go out to 6 lags) in the photon correlation calculation. We fear it may induce more complications to link the two times, as in that 20 ms window we are basically going through every photon and finding instances of where there is a photon on the other channel within the last 5-10 laser pulses. i.e. the fact that the total window is 800k pulses doesn't really matter as we never correlate anywhere near that long.

Reviewer 3:

We thank the reviewer for their comments on the high quality of our work and its suitability for Nature Communications. The reviewer suggests some aspects that should be discussed or addressed before publication:

“1. The authors studied single PFO chains with a molecular weight of $M_w \sim 200$ kDa, which most likely supports both interchain and intrachain interactions. It would be interesting to differentiate interchain and intrachain excitons via studying a relatively lower M_w single polymer chain (e.g., tens of kDa or lower) and/or a single aggregate of low- M_w chains. A relatively clearer picture of exciton interaction physics can be obtained, which could offer some potential strategies in materials or device

perspectives to achieve high exciton densities for brighter polymer-based OLEDs or lasers.”

We appreciate the reviewers' comments and agree that this is an interesting area to explore. Unfortunately, it is less practicable to do this with new measurements here due to the time required. To give some guidance, the data in Figure 3 (~5000 chains) took on the order of 3 months to acquire. If we used recovery GPC to create even just 4 separate Mw fractions that would require us to spend another full year measuring. Even this ignores the fact that for the lower weight fractions longer would be needed due to lower count rates. However, the reviewer's comments are insightful, and so we have taken them as inspiration to analyse our existing data in new ways that we did not envisage originally, undertaking chain brightness dependent analysis of the annihilation behaviour. To explore the reviewer's comment, we are going to have to rely on the assumption that chain brightness scales with chain length. We have no way to avoid this assumption, and accept that other effects, e.g. conformational heterogeneity, can also play a role, but in general terms this will be true, i.e. the longer the chain is the more excitation light it will absorb, and thus the more it will emit. Taking this assumption as true, it gives us an analogue to chain length to use and compare with.

Firstly, we have taken the 5012 chains and extracted their maximum brightness per trace (in any 20 ms bin) and plotted as a histogram, giving us a possible equivalent to the chain length. This follows an approximate exponential distribution and so have plotted on a log scale, with an inset zoomed in on the 0-75 kHz range:

We note this is consistent with the new GPC results we have obtained, as shown above and in SI, and suggestive that the main molecular weight peak (M_n of 15100 g

mol^{-1} and M_w of 30400 g mol^{-1}) is likely represented in the exponential distribution observed here, while the small high molecular weight peak (M_n of $471000 \text{ g mol}^{-1}$ and M_w of $555000 \text{ g mol}^{-1}$) will be chains that are much brighter (75+ kHz).

Now we can split this into four regions, 0-25, 25-50, 50-75 and 75+ kHz and analyse the time-resolved antibunching behaviour as per Figure 3d for each of them, but now for different effective chain lengths (and for the blue-, yellow- and red site colour ratios). We do not enforce any minimum count rate as we did in Figure 3d (for the fractions chosen are themselves the natural filters on count rates):

The results are a little difficult to discuss with strong confidence owing to signal-to-noise limitations, however overall trends can be discerned, as indicated with the black arrows on each plot showing the general change in the number of independent emitting sites as the chains get longer.

The emission sites denoted blue and yellow, as per the main text, both show the same behaviour and trend, confirming that these two are very similar. It is observed that the brighter/longer the chain is, the faster the annihilation and the lower the number of final emitting sites on the chain once annihilation is completed. This is an interesting result and suggests that smaller chains have fewer opportunities for excitons to diffuse and meet each other to annihilate, this is likely because diffusion is limited to along single chains with 1D motion, and thus large orientational changes in chain segments, defects or local disorder can restrict motion. In contrast, longer chains are observed to have faster/more efficient annihilation, this is a consequence of the potential for greater pseudo-interchain hopping as chains fold back on themselves giving increased opportunities for diffusion.

In contrast, the red low-energy emission sites show the opposite trend, where short chains reach a lower total number of emitting sites, and long chains a higher value. This result is consistent with a view of these low-energy sites being positioned at a low density on the chain. When the chains are short the number of these sites is low, and this increases as chains get bigger. The low number of these sites enforces slow annihilation, consistent with our original discussion/assignment in the main text. While in the main text we indicate that pseudo-interchain interactions are likely how these red sites annihilate with each other, the density of these sites is low enough that the increased chain length does not appear to enhance the potential for annihilation, as an interchain speedup will only be observed if such a bridge forms at/near the red sites (in contrast to the blue sites, where any interchain bridging can be exploited for faster annihilation as they sit on all ordinary fluorene units). However, we are cautious on

overinterpreting the intensity-dependent results we have extracted here, as all traces are clearly noisy, even if overall trends are visible.

We have added these plots to SI along with explanatory text, and have added text to the main manuscript to describe/discuss this brightness dependence as an analogue for chain length, with the caveat that future studies to explore this more precisely would be interesting owing to the limited signal-to-noise we have here (P15):

“One way to explore this is by measuring with a range of defined chain lengths and comparing the annihilation behaviour. Unfortunately, such a study was beyond the scope of this work, however, we have explored this indirectly by using chain brightness as an analogue for length (see SI for such extracted plots). We find that, as suggested immediately above, pseudo-interchain coupling is more likely in brighter (longer) chains, and thus they show more efficient annihilation for blue sites, while weaker (shorter) chains show less annihilation, as opportunities for excitons to hop to different chain segments are hindered.”

“2. Did the authors check the polymer matrix (e.g., dielectric) effect on single PFO chain behavior? The matrix environment strongly affects the EEA in terms of exciton diffusion, exciton binding energy, and energy transfer processes. Some discussion or investigation on this aspect should be added.”

We did some early exploratory measurements with a cyclic olefin polymer, which has different polarity than PMMA and saw no differences in simple single chain photophysical parameters (e.g. lifetime, conventional antibunching) and so did not pursue its usage further. We are happy to discuss possible effects of the matrix in the manuscript and have added text (P15) to do this:

“Control of the conformation of the chains would also be a good way to change the degree of interchain coupling, and this could potentially be achieved at the single molecule level by changing the host matrix.⁵² We conducted some initial early trials with a non-polar cyclic olefin host material; however we found no obvious difference in basic photophysical properties so did not pursue this further. If a host was found to strongly alter the PFO chain conformations and local environment (e.g. dielectric constant) then this would be a powerful way to explore interchain annihilation in further detail.”

“3. What is the noise level of time-gated antibunching data? Was the noise corrected for the presented data?”

The definition of noise for time-resolved (or time-integrated) antibunching data is entirely down to the ratio between the occurrences on the central (N_c) and lateral (N_l) bins. One standard deviation is where we derive the error margins we quote on the number of independent emitting sites in Figure 2 (i.e. N_c/N_l varies in the range from $N_c+1\sigma N_c / N_l-1\sigma N_l$ through to $N_c-1\sigma N_c / N_l+1\sigma N_l$). For the lateral bins we have a large sample size (typically we calculate to 50 lags both sides) but there is an issue in all correlation measurements like these which is that for the central bin we have only one sample, and thus typically use the N_l standard deviation on the central bin as well for want of a better solution.

Consequently, we can construct a 1σ confidence interval-derived \pm error for the time-resolved data shown in Figure 3d, and we have added this in SI, along with some explanatory text:

It is observed that the difference between the blue/yellow and red decays are fully separable and lie outside the error ranges of each other for most of the time range. We are amenable to using this rendering in the main text Figure 3d if the reviewer supports this but in the first instance judged that it impinges upon legibility a little for the main text presentation, so have added it to SI only. We have added a signpost in the main text to enable the reader to find it in the SI (P13):

“To confirm separability between the decay in the number of emitters for blue/yellow and red sites we can calculate the error on each to provide ranges that they lie within, this is shown in Supplementary Information.”

We do not correct for noise in any way in the presented data, i.e. there is no background subtraction or noise gating. For pure photon counting this is difficult to do,

as we deal with individual binary events when looking for correlation pairs. For each correlation pair there is almost nothing to say whether either photon is from photoluminescence or noise (we could exclude photons arriving before the laser pulse, but these are very small in number and have almost no effect on correlation counts). That, amongst other reasons, is why we use detectors such as SPADs with the lowest possible dark counts. As noted above we have applied a gentle intensity floor to the 20 ms slices we analyse, taking only those that have ≥ 200 photons detected in them. The effect of this is minor but removes the risk of including photons when the chains are dark and only noise is detected (which will show no antibunching and thus lift N_d/N_i values). Finally, we note that there was a small error in how we applied this ≥ 200 count filter in the code that was used to produce the plot in the original submission's Figure 3d. We have fixed this and updated Figure 3d and all plots where that data is used so the traces are fully correct. We note the error made almost no difference to any of the traces and no difference to any conclusions.

Reviewer 4:

We thank the reviewer for the time they have taken to read and comment on our work and hope the process has been of benefit for their future reviewing activities.